# Beyond Temporal Credit Assignment in Reinforcement Learning

**Sephora Madjiheurem, Kimberly L. Stachenfeld, Peter W. Battaglia & Jessica B. Hamrick**
DeepMind, London, UK
{sephoram, stachenfeld, peterbattaglia, jhamrick}@deepmind.com

## Abstract

In reinforcement learning, traditional value-based methods rely heavily on time as the main proxy for propagating information across the state space. This often results in slow learning and does not scale to large and complex environments. Here, we propose to leverage prior information about the structure of the the environment to assign credit non-temporally to improve learning efficiency. Specifically, we introduce the concept of *structural neighbours*, which are sets of states with similar semantic structures and which have equivalent values under the optimal policy. We augment traditional value-based RL methods (TD(0), Dyna and Dueling DQN) with a learning mechanism based on structural neighbours. Our empirical results show that by incorporating structural updates, learning efficiency can be greatly improved on a variety of environments ranging from simple tabular grid worlds to those which require function approximation, including the complex and high-dimensional game of Solitaire.

## 1 Introduction

Reinforcement learning (RL) has seen immense interest over the last decade, demonstrating that given enough engineering and compute, RL agents can solve just about any environment they are presented with (Silver et al., 2016; Berner et al., 2019; Baker et al., 2020; Akkaya et al., 2019; Vinyals et al., 2019; Schmid et al., 2021). Yet despite these successes, this approach only scales so far: it is simply not feasible to solve every problem from scratch by brute force. As a field, it is crucial to develop methods which can more effectively leverage prior knowledge to speed up learning and make RL more tractable on a wider variety of problems.

Here, we question an assumption made by traditional RL methods that limits their learning speed, which is that *time* ought to be the primary dimension along which to propagate information about reward. Temporal credit assignment can be incredibly slow because at each iteration of the learning algorithm, only a small set of states—those encountered during the rewarding trajectory—are updated. This issue is compounded in environments with large state spaces and sparse rewards (van Hasselt et al., 2021). As such, we ask the question: can we speed up learning by additionally assigning credit *non-temporally*? It is straightforward to see that many environ-

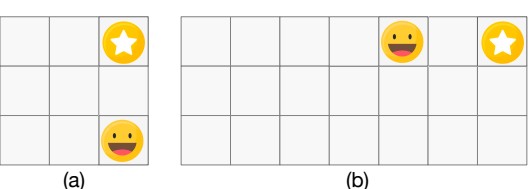

Figure 1: A motivating example for structural updates. If we are in a world in which rewards always exist in the top right corner of a room, then after experiencing a rewarding trajectory with the state in (a), we should immediately be able to transfer that knowledge to the state in (b).

ments exhibit non-temporal structures, for example: states that are all equidistant to a goal (e.g. Figure 1); states that are observed under different environmental conditions (e.g. lighting, viewpoint, etc.); and states that share other structural similarities (e.g., cooking the same meal on two different stoves, or sorting laundry into the same piles but in different places on the floor). We believe that these non-temporal structures can be effectively leveraged to accelerate learning.

In this paper, we propose treating states with non-temporal similarities as *structural neighbours* and propagate reward information to them as well as to temporal neighbours. We make the particular assumption that, under the optimal policy, structural neighbours are states which have equivalent values; thus, any update to the value of one state should be applicable to all of its structural neighbours. We show that structural updates can be easily incorporated into any value-based RL algorithm, including Dyna (Sutton et al., 1999), TD(0) (Sutton & Barto, 2018), and Dueling DQN (Wang et al., 2015). Using structural updates, we demonstrate vastly improved learning efficiency in tabular grid worlds, a miniature tabular version of Solitaire, and the full version of Solitaire with function approximation. We close with a discussion of our results and some thoughts on how such prior structural knowledge might be extracted from pre-trained systems such as LLMs.

## 2 RELATED WORK

**Equivariant RL**  This work is by no means the first to identify that some RL environments contain non-temporal symmetries relevant to the learning of the value function and policy. Work by van der Pol et al. (2020) introduce a way to build in this prior knowledge into policy and value networks. Mondal et al. (2022) propose to learn symmetry transformations of the state-action pairs alongside representations that are equivariant to the agent's actions. However, these works assume the existence of discoverable action (or state-action) mappings. Our approach generalises this concept by proposing a way of using structural information present in the state space without requiring similar knowledge about the actions. More similar to our approach would be to use an equivariant state value function; however, structural updates allow us to relax the strict equivariance assumption while still accelerating learning.

**Long-term credit assignment**  The general problem of credit assignment in RL has, similarly, long been of interest. Recently, Raposo et al. (2021) proposed a state-associative learning mechanism which models the contribution of past states to the current reward. This model then allows for credit to be assigned to states that are arbitrarily distant from future rewards. van Hasselt et al. (2021) introduced the notion of expected eligibility traces, which extend instantaneous eligibility traces by also capturing counterfactual past states that could have led to the same outcome. In this way, the rewarding information can be propagated more broadly through the state space. Unlike our work, these methods only address a limitation of temporal credit assignment, and do not consider assigning credit to states that might be temporally completely unrelated to the current reward.

**Experience replay and memory**  In RL, the use of a replay buffer allows us to learn from past experience that are arbitrary far back in time. Work by Schaul et al. (2016) and Andrychowicz et al. (2017) introduce mechanisms for selecting past experiences that are most relevant for policy learning. Pritzel et al. (2017) propose a representation of its experience which possesses features of episodic memory and allows to retain successful strategies as soon as they are experienced. Goyal et al. (2022) propose to train a retrieval process to map past experiences to optimal behavior. While experience replay and memory are helpful to combat some of the limitations of traditional online temporal learning, they still only allow us to update states that were visited at least once. Our method uses structural knowledge to update states that might not be present in the agent's memory. Put another way, existing methods support more rapid *consolidation* of prior experience, while our method supports *generalization* of experience.

**Other ways of leveraging prior knowledge**  There are many ways to incorporate prior knowledge into RL agents, including architectural priors (Wang et al., 2018; Almasan et al., 2019), model-based RL (Schrittwieser et al., 2019), pre-training (Agarwal et al., 2021; Higgins et al., 2017; Parisi et al., 2022), and recently, interfacing with large language models (LLMs) (Huang et al., 2022; Fan et al., 2022). All of these methods aim to leverage better representations of the task while leaving the core reinforcement learning algorithm unchanged. Here, we consider the inverse approach: incorporating prior knowledge into the reinforcement learning or credit assignment algorithm itself.

## 3  BACKGROUND AND MOTIVATION

We define a discrete MDP by the tuple $M = (\mathcal{S}, \mathcal{A}, P, R)$, where $\mathcal{S}$ is a finite set of discrete states, $\mathcal{A}$ a finite set of actions, $P$ describes the transition model (with $P(s, a, s')$ giving the probability of moving from state $s$ to $s'$ given action $a$), and $R$ describes the reward function (with $R(s, a)$ giving the immediate reward from taking action $a$ in state $s$). Solving this MDP involves finding a policy $\pi : \mathcal{S} \mapsto \mathcal{A}$ which maximizes the expected discounted sum of rewards.

A common class of methods for solving MDPs are known as *value-based* methods because they estimate a value function $V_\pi : \mathcal{S} \mapsto \mathbb{R}$ describing the expected long-term discounted sum of rewards observed by the agent in any given state $s$ when following policy $\pi$. In value-based methods, the goal is to find the optimal value function $V^*$ satisfying the Bellman optimality equation (and corresponding to the optimal policy $\pi^*$):

$$V^*(s) = \max_a \left( R(s, a) + \gamma \sum_{s' \in S} P(s, a, s') V^*(s') \right). \tag{1}$$

As described in the following section, different methods take different approaches to finding $V^*$.

### 3.1  VALUE-BASED REINFORCEMENT LEARNING

**Value Iteration**    In tabular settings where the reward and transition functions are known, the value function in Equation 1 can be computed by dynamic programming, iteratively evaluating the value functions for all states:

$$V(s) \leftarrow \max_a \left( R(s, a) + \gamma \sum_{s'} P(s, a, s') V(s') \right), \forall s \in \mathcal{S} \tag{2}$$

This is the value iteration algorithm and is shown to converge to the optimal value function with an infinite number of steps (Sutton & Barto, 2018).

**Dyna**    Approximate dynamic programming methods such as Dyna (Sutton, 1990) have been developed to solve RL problems when the reward and/or transition functions are unknown. These methods use past experiences to learn a model of the environment (i.e. approximations of reward function $R$ and/or the transition dynamic $P$), and use this model to update the state value estimates.

**Temporal-Difference learning**    Temporal-Difference (TD) learning learning provides a way of learning the value function from experience, in the absence of a model and without requiring to sample full trajectories. TD learning achieves such *online learning* via bootstrapping, by updating the current value estimates towards a predicted return based on the current value estimates. The TD prediction update is as follows:

$$V(s_t) \leftarrow V(s_t) + \alpha \big[ \underbrace{R_{t+1} + \gamma V(s_{t+1}) - V(s_t)}_{\text{TD-error}} \big]. \tag{3}$$

To recover the optimal policy, we may alternately seek to learn a function $Q : \mathcal{S} \times \mathcal{A} \mapsto \mathbb{R}$, which describes the value at a given state when taking the specific action. The corresponding TD learning method for control is known as Q-learning (Sutton & Barto, 2018), and the update at time $t$ is:

$$Q(s_t, a_t) \leftarrow Q(s_t, a_t) + \alpha \big[ R_{t+1} + \gamma \max_a Q(s_{t+1}, a) - Q(s_t, a_t) \big] \tag{4}$$

**Function approximation**    In practice, as the state and action spaces grow, tabular methods become infeasible and function approximation is often used to mitigate the issue. For instance, instead of learning a direct mapping $V : \mathcal{S} \mapsto \mathbb{R}$, a parameterised function is learned instead: $V_\theta(s) \approx V(s)$, with $\theta$ the learned parameters. Similarly, the $Q$ function can be approximated by a parameterised function, such as a deep neural network (Mnih et al., 2015; Wang et al., 2015).

### 3.2  LIMITATIONS OF TEMPORAL CREDIT ASSIGNMENT

The RL methods described in the previous section rely on time as proxy to propagate reward information across the state space. In large state spaces with sparse reward, this results in slow information propagation: when a rewarding event is observed, only previous states in the current trajectory

will be updated. However, the experience of visiting a rewarding state might tell us more than just that visited states led to a reward. We might also be able to leverage prior knowledge to infer that *similar* states could lead to rewards, too.

As a motivating example, consider the environment in Figure 1. In this environment, the agent finds itself in a room of a random size. In every room, there exists a reward in the top right corner. In a traditional RL approach, the agent must experience many different rooms of different sizes in order to eventually learn that states (a) and (b) in the figure have the same value (because the agent is equally far away from the reward). However, if we are told explicitly that the rewards are always in the same place, then this knowledge should be exploitable: after experiencing a rewarding trajectory containing (a), the agent ought to be able to use this information to know what to do when encountering (b) even when the reward in the top right corner is never directly observed.

In theory, function approximation should help with this issue of assigning credit to similar states: states with similar feature vectors should also see their values updated. However, using function approximation in this way to speed up learning—particularly at the beginning of training—poses a chicken-and-egg problem. To help with assigning credit, we already need to have a good representation, but to get a good representation, we need high-quality data across a diversity of states in regions of non-zero reward, which is usually only available after we have already had some success with assigning credit!

Our intuition is that we should explicitly consider a *set* of related or similar states, rather than implicitly considering them through function approximation. In particular, our approach is motivated by the idea that each encountered state should lead to multiple updates: one for each related state. In our motivating example, we would not simply update the state shown in Figure 1a, but all states in which the agent is equidistant to the top right corner (regardless of room size).

The idea of encoding prior knowledge as sets of related states is highly general and allows us to capture complex relationships between different states that might not be easily expressed using alternate schemes. For example, in Figure 1, we could alternately encode prior information about the rewarding corner

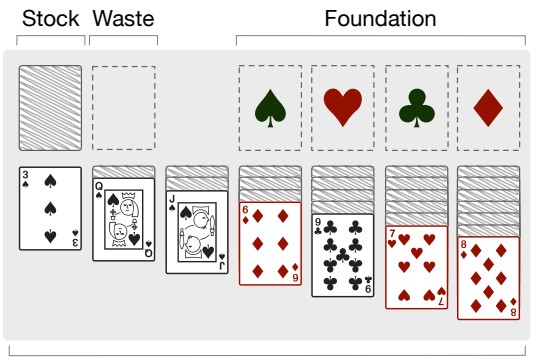

Figure 2: An example initial Solitaire game. The goal is to move all cards to the *foundation* such that they are separated by suit and sorted by rank, with aces on the bottom and kings on the top. To play, cards may be drawn from the *stock* pile and either placed on top of another visible card, or on the *waste* pile. In the *tableau*, cards may be placed under the constraint that visible cards must alternate colors and be placed in increasing order. Full piles of visible cards may also be moved at the same time. If a hidden card in the tableau becomes the top card, then it may be revealed. Valid moves in this board would be to move the 8 of Diamonds on top of the 9 of Clubs, or to draw a card from the stock pile. After moving the 8 of Diamonds, the hidden card beneath it may be revealed.

using a given reward function, or using an egocentric representation. Yet consider more complicated settings, such as card games like Solitaire (Figure 2). Experienced players know that, regardless of the particular rules, various relationships between the cards exist across games: cards may be ordered by suit, they can be placed in piles, they can be face up or face down, etc. Moreover, the rules are often equivariant with respect to particular ranks or suits (e.g., if you can place a heart on top of a heart, then it is likely that you can place a spade on top of a spade). These types of relationships are not easily captured by the reward function because they are about the rules of the game more generally. However, they *can* be captured through the idea of sets of related states.

# 4  STRUCTURAL REINFORCEMENT LEARNING

We are now ready to formally define the notion of structural neighbours. We will also explain how to use this concept to build structural updates that can be plugged into traditional RL algorithms to improve learning efficiency.

**Definition 1.** *A **structural transformation** is a transformation $T_k : \mathcal{S} \mapsto \mathcal{S}$ such that $V(s) = V(T_k(s))$, in other words, a state transformation to which the value function is invariant.*

**Definition 2.** *The **structural neighbours** of state $s_i$ are a set of size $K$ denoted by $\mathcal{N}_K(s_i)$, and are constructed by applying by $K$ different structural transformations to $s_i$, i.e. $\mathcal{N}_K(s_i) = \{T_1(s_i), T_2(s_i), \ldots, T_K(s_i)\}$.*

Note that the notion of structural neighbours relates to state value similarity and does not imply state-action similarity (i.e. $Q(s_i, a)$ need not to be equal to $Q(s_j, a)$ for any $a \in \mathcal{A}$). In the remaining of the paper, we assume that we have access to a pre-computed neighbourhood function $\mathcal{N}_K$. For ease of reading, we omit the subscript and write $\mathcal{N}$. We will return to the question of where these neighbourhood functions might come from in the discussion.

**Structural Dyna**   We consider first the simple scenario of solving the value prediction problem in tabular environments with known transition function and unknown reward function. We propose to use a type of Dyna architecture (Sutton, 1990) which alternates between (1) learning a reward model $\hat{R}$ from experienced trajectories with a regression loss, and (2) using this reward model to update value estimates according to the value iteration update Equation 2. We propose to increase the propagation of reward information beyond sampled states by incorporating following structural updates to the Dyna algorithm:

$$\hat{R}(s_j) = \eta \hat{R}(s_t) + (1 - \eta)\hat{R}(s_j), \forall s_j \in \mathcal{N}(s_t) \tag{5}$$

where $\eta \in [0, 1]$ is a mixing factor such that $\eta = 0$ corresponds to no structural updates, and $\eta = 1$ corresponds to no temporal updates. We call this algorithm Structural Dyna. Note that if whenever $R(s_i) \neq 0$ we have that $s_i$ is a terminating state, then it follows that if $s_i$ and $s_j$ are structural neighbours, $R(s_i) = R(s_j)$ and we can set $\eta = 1$. Structural Dyna can be extended to non-tabular environments by learning a parametrised reward function $R_\theta$, and updating the parameters via gradient descent methods, where at time $t$ the regression targets are $\eta R_\theta(s_t) + (1 - \eta)R_\theta(s_j)$ for all $s_j \in \mathcal{N}(s_t)$.

**Structural TD learning**   Structural prior knowledge about the environment can also be used in model-free RL, where we directly learn the value function without having access to, or learning, a model of the environment. We propose to augment TD methods with a structural update immediately following the temporal update. The idea is to propagate at each time step the estimate of the value at the current state $s_t$ onto its structural neighbours using the following updates:

$$V(s_t) \leftarrow V(s_t) + \alpha\big[R_{t+1} + \gamma V(s_{t+1}) - V(s_t)\big] \qquad \text{(temporal update)} \tag{6}$$

$$V(s_j) \leftarrow \eta V(s_t) + (1 - \eta)V(s_j), \ \forall s_j \in \mathcal{N}(s_t) \qquad \text{(structural updates)} \tag{7}$$

where $\eta \in [0, 1]$ is a mixing factor where, as before $\eta = 0$ corresponds to no structural updates and $\eta = 1$ corresponds to fully overwriting temporal values with those from structural neighbors.

In the function approximation case where the value function is approximated by a parametric function $V_\theta$, the structural updates consists in taking a gradient step with respect to the parameters $\theta$ towards minimising the following structural difference loss:

$$\ell_{SD}(s_t) = \frac{1}{|\mathcal{N}(s_t)|} \sum_{s_j \in \mathcal{N}(s_t)} \big(\eta V_\theta(s_t) + (1 - \eta)V_\theta(s_j) - V_\theta(s_j)\big)^2 \tag{8}$$

We call this augmented version of TD learning Structural TD learning.

**Structural Dueling DQN**   We now demonstrate how the structural updates can be integrated into a deep RL algorithm for control. Value based methods for control seek to find the optimal state value function by updating estimates according to Equation 4. Note however that we do not impose

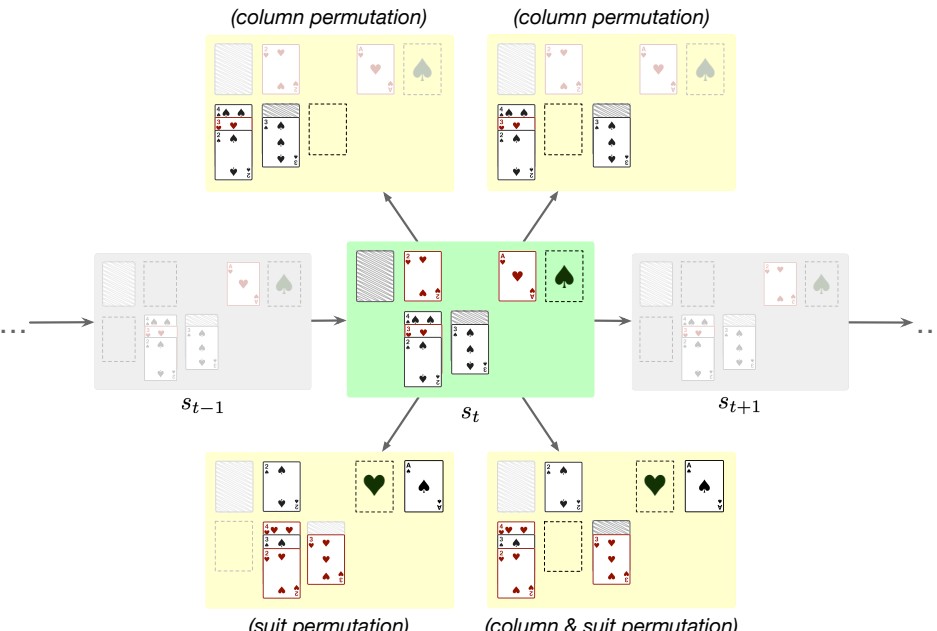

Figure 3: Illustration of the different types of state neighbours in Mini Solitaire. The temporal neighbours are shown in gray boxes, while the structural neighbours are shown in yellow boxes. The values of the temporally connected states are related according to Bellman's Equation 1. The values of the structural neighbours are exactly equivalent.

any a priori knowledge about the state-action structural mappings (meaning that $s_j \in \mathcal{N}(s_i)$ does not necessarily imply that $Q(s_i, a) = Q(s_j, a)$). This prevents us from applying structural updates directly to the $Q$-function. We therefore adopt the following decomposition of the $Q$-function: $Q_\theta(s, a) = V_\theta(s) + A_\theta(s, a)$. Here, $A$ is the *advantage function*, describing the importance of an action $a$ at state $s$. Deep RL algorithms using this decomposition to learn the $Q$-function are known as Dueling Deep Q-Networks (DQNs) (Wang et al., 2015). With this decomposition, we can apply our structural updates according to Equation 6 to the value head of the network only (back-propagating through the shared torso), while the rest of the network is updated only with temporal updates. In this way, we allow for value information through propagate across structural neighbours without influencing the effect of the actions. We call this algorithm Structural Dueling DQN.

**Proof of convergence** Given our definition of structural updates as being a weighted combination of temporally-derived values and values from structural neighbors, we can show that using structural updates will under certain assumptions provably converge to the optimal value function. See Appendix D for details.

## 5 RESULTS

### 5.1 ENVIRONMENTS

**Rooms** We consider a modification of the four rooms domain (Sutton et al., 1999). This environment is a grid world divided into four rooms connected by doorways. The whole grid is of size 10x10, with each room being 5x5. There are four actions at each state: move up, move right, move down and move left. There is a single reward per room, always located at the same location within a room (e.g. always in the upper right corner, or always in lower right corner, etc.). The agent is initially placed randomly at a non rewarding state, and has to navigate the environment until a reward is found. In this environment, due to the fact that rewards are placed at the same location within rooms, a structural transformation applied to state $s_i$ in a given room consist in taking the state at the same location location as $s_i$ in any other room. We construct a structural neighbourhood

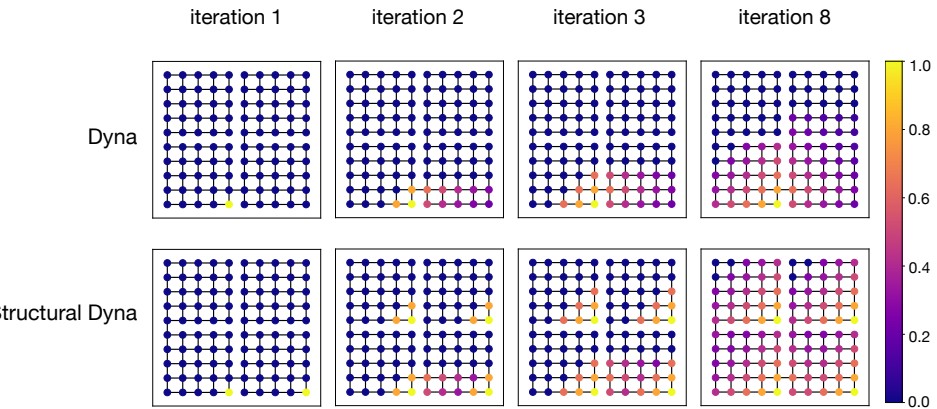

Figure 4: State values at different iterations of Dyna (top row) and Structural Dyna (bottom row) on Rooms environment. Yellow indicates a high value, while purple indicates a low value. In this setting, we know that rewards always occur in the same corners of every room. After the agent observes a reward at the lower right corner, Structural Dyna is able to use this prior knowledge to propagate values to all rooms, while regular Dyna is not able to do so.

of $s_i$ by considering all three other states at the identical location in the other rooms. Note that this environment is not strictly equivariant: the passages between rooms are not in the same places, therefore the state values near those passages will not be strictly equivalent under the optimal value function. This environment therefore serves as a test of the robustness of structural updates when its underlying assumptions are violated.

**Solitaire**    Solitaire (Figure 2) is a one player game in which cards from a standard deck are initially shuffled and then laid out in a specific arrangement across a board. The board is arranged in four categories: the tableau, the stock, the waste, and the foundation. The goal is to sort all the cards by suit in increasing rank ordering into four piles in the foundation (one for each suit) by sequentially moving and revealing cards. The rules of the game are detailed in Appendix A. We model the MDP environment with a symbolic implementation of Solitaire, in which each card is represented by a unique encoding and a state observation is a concatenation of the visible cards on the board. See Appendix A for a description of the state and action spaces and the rewards. In this game, permuting suits of similar colour does not affect the state value: a state in which the diamonds are replaced with hearts should have the same value as the original state with diamonds. Thus, we define the structural neighbourhood of state $s_i$ to be all states with equivalent layouts, but with suits permuted. For each state, there are 5 such valid suit permutations: $\{(\heartsuit \leftrightarrow \diamondsuit), (\clubsuit \leftrightarrow \spadesuit), (\heartsuit \leftrightarrow \diamondsuit \text{ and } \clubsuit \leftrightarrow \spadesuit), (\heartsuit \leftrightarrow \clubsuit \text{ and } \diamondsuit \leftrightarrow \spadesuit), (\heartsuit \leftrightarrow \spadesuit \text{ and } \diamondsuit \leftrightarrow \clubsuit)\}$.

**Mini Solitaire**    This game is played with a deck of eight standard playing cards consisting of two suits of different colour (Spades and Hearts) and four ranks (Ace, 1, 2, 3). The rules of the game are the same as the full version of Solitaire, but with reduced number of piles on the board (two foundation piles and three tableau piles). Initially, the first tableau pile has 1 card, the second had 2 cards and the third has 3, and only the last card of each tableau the pile is visible. In Mini Solitaire, there are 166124 valid configurations of the cards and there are 41 possible actions. The reward function is similar as that of Solitaire (Appendix A). In Mini Solitaire, since we only have 2 suits, there is only one valid suit permutation. We therefore consider another kind of transformation: tableau columns permutations. Indeed, a state with identical layout but where the columns in the tableau are permuted preserves state equivalence (see Figure 3 for an illustration). We construct the state neighbourhoods $\mathcal{N}_{11}(\cdot)$ by applying 11 transformations (permutation of the suits, 5 columns permutation of the order of the tableau piles, and 5 columns permutation of the order of the tableau piles with permuted suits).

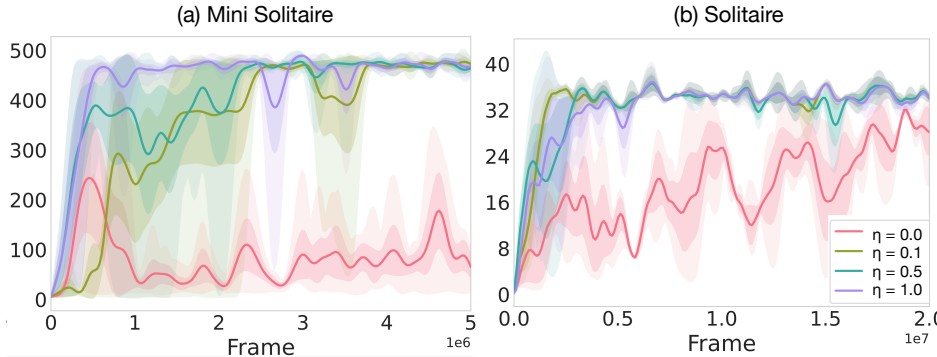

Figure 5: Structural Dueling DQN on (a) Mini Solitaire and (b) Solitaire. In both plots, $\eta = 0$ corresponds to no structural updates, while increasing values of $\eta$ correspond to increasing weight given to structural information. The $x$-axes show the number of frames, and the $y$-axes shows total return. The bold lines are average over 5 random seeds, and the shaded areas are the standard errors. In all cases, incorporating structural information enables the agent to learn much more quickly and achieve higher rewards.

## 5.2 EXPERIMENTS

**Structural Dyna** To validate our intuitions about structural neighbours, we run a simple experiment on the Rooms environment in which we consider only a single trajectory encountering a single rewarding corner. Figure 4 shows the value function at different stages of learning for both Dyna and Structural Dyna. As we would expect, Structural Dyna allows the state values to accurately propagate beyond temporally connected states. This is true even though the optimal value function of this environment does not exhibit strict equivariance, as discussed in Section 5.1. In contrast, Dyna can only propagate seen rewards through time.

**Structural Dueling DQN** We also evaluate Structural Dueling DQN on Mini Solitaire and Solitaire. All models have the same architecture: a torso state encoder MLP network consisting of 3 layers of 512 units, a value head and an attention head of 512 units and a final layer of size 161 (the number of possible actions). The network is trained with a batch size of 32, using experience replay and a target network (Mnih et al., 2015; Wang et al., 2015). We update the target network every 1000 training steps. We adopt a $\epsilon$-greedy strategy, starting with $\epsilon = 1.0$ and decaying over the first 2% of the total number of training steps to $\epsilon = 0.01$. The learning rate for the temporal updates is 0.003, and the learning rate for the structural updates is 0.01.

Figure 5 shows the results. In both environments, the benefit of having structural updates in addition to temporal updates is apparent: Structural Dueling DQN ($\eta > 0$) reaches a high total return in considerably fewer training steps. Without structural updates ($\eta = 0$), we find that the agent is quite slow and did not even converge in the allotted training time.

## 6 CONCLUSION

In this paper, we introduced the concept of non-temporal credit assignment via *structural neighbours*: sets of states with structural equivalencies such that their values under the optimal policy are the same. We showed how structural updates can be easily added to existing value-based RL algorithms, and demonstrated empirically how structural updates can accelerate learning across three environments.

In all our experiments, we assumed access to a structural transformation function that maps from states to structural neighbours. This is a big assumption, and such a function might not be readily available, especially in non-symbolic environments. However, with recent advances in large models, we are optimistic that constructing structural transformations automatically may not be that far out of reach. For example, recent models for large-scale retrieval are able to leverage large bodies of experience to retrieve not just to retrieve similar states with superficial differences, but with structural

similarities too (Humphreys et al., 2022) and thus could potentially be used to retrieve our structural neighbourhoods, too. Another possibility could build on foundation models: for example, we could ask a large language model like ChatGPT (OpenAI, 2022) to describe similar states to the current state, and then, conditioned on its response, ask a diffusion-based image editing model (Kawar et al., 2022) to transform the current state to structural neighbours. We consider such methods for automatic structural transformations to be a key area of future work.

Overall, we believe that the speed of RL is limited by an over-reliance on temporal credit assignment. By developing new approaches to distribute credit *non-temporally*, we take a step towards more flexible, adaptive learning systems which rapidly consolidate and generalize new experiences.

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

# A    SOLITAIRE

The board is arranged in four categories: the tableau, the stock, the waste, and the foundation. The tableau is made of seven piles. At the start of the game, the first pile has one card, the second pile has two cards, the third pile has three cards, and so on until the seventh pile. All cards except the last of each pile is face down. The remaining cards are faced down in the stock pile. For a card on the top of a pile, the player can either move it (if it is face up) or reveal it (if it is face down). The rule for moving a card is as follows: Cards from the tableau or the waste pile can either be moved onto a different pile on the tableau in increasing rank order and alternating suit colour, or moved onto the foundation in increasing rank order of matching suit. Cards from the stock pile can be moved to the same manner or discarded into the waste pile. Cards from the foundation can be moved back into the tableau, respecting the increasing rank order and alternating suit colour. Visible cards on the tableau from the same pile can also be moved all at once and put in a different pile in the tableau, respecting the ranking and alternating suit colour rules (note that our Solitaire implementation does not allow for piles to be split: we can either move the last card in a tableau pile or all the visible cards).

There are 97 cards position on the board at a given time (accounting for visible cards, hidden cards, and empty slots where cards could go), 6 suits (Hearts, Spades, Diamonds, Clubs, hidden, empty) and 15 ranks (Ace, 1, 2, ..., Jack, Queen, King, hidden and empty). A state is thus represented as binary vector of size (97, 21). An action is defined by three numbers $(a_1, a_2, a_3)$, $a_1$ represents the location of the card or cards to be moved or revealed, $a_2$ is the destination location, and $a_2$ indicates whether to move one card or the whole pile. In practice, we only consider valid actions and convert this tuple into an integer. There are 161 valid actions in total. The agent receives a reward of $+10$ for revealing a new hidden card from the tableau, $+20$ for stacking a new card into a foundation pile, $-20$ for removing a card from the foundation, and $500$ for putting the last card from the tableau onto the foundation.

## B PSEUDOCODE

---
**Algorithm 1** Structural Dyna
---
**Input:**
$\pi$: exploratory policy,
$P$: transition dynamic,
$N$: number of episodes,
$M$: Number of iterations
**Initialise:** $V(s) = 0$, $R(s) = 0 \, \forall s$, $s \sim \mathcal{S}$, $r = 0$, $a \sim \pi(s)$
**Output:** estimated optimal value function $V$
**for** N episodes **do**
    **while** $s$ is not terminal **do**
        $s, r = step(a)$
        $R(s) = r$
        $a \sim \pi$
    **end while**
**end for**
**for** M iterations **do**
    **for** each state $s$ **do**
        $V(s) = max_a(R(s,a) + \gamma \sum_{s'} P(s,a,s')V(s))$
        **if** $R(s) \neq 0$ **then**
            **for** $n \in \mathcal{N}(s)$ **do**
                $R(n) = \eta R(s) + (1 - \eta)R(n)$
            **end for**
        **end if**
    **end for**
**end for**

---
**Algorithm 2** Structural Tabular TD($\lambda$)
---
1: initialise $v(s) = 0 \; \forall s$
2: **for** $M$ episodes **do**
3:     initialise $\boldsymbol{e} = \boldsymbol{0}$
4:     observe initial state $s$
5:     **repeat** for each step in episode $m$
6:         generate $R$ and $s_{t+1}$
7:         $\delta_t \leftarrow R_t + \gamma v(s_{t+1})$
8:         $\boldsymbol{e} \leftarrow \gamma \lambda \boldsymbol{e} + \phi(s)$
9:         $v(s_t) \leftarrow v(s_t) + \alpha \delta \boldsymbol{e}$
10:        **for** $s_k$ in $\mathcal{N}(s_t)$ **do**
11:           $v(s_k) \leftarrow (1 - \eta)v(s_t) + \eta v(s_k)$
12:        **end for**
13:     **until** $s$ is terminal
14: **end for**
15: **Return w**

---

**Algorithm 3** Structural TD($\lambda$) with function approximation

1: initialise $\mathbf{w}$
2: **for** $M$ episodes **do**
3:     initialise $\boldsymbol{e} = \mathbf{0}$
4:     observe initial state $s$
5:     **repeat** for each step in episode $m$
6:         generate $R$ and $s_{t+1}$
7:         $\delta_t \leftarrow R_t + \gamma v_{\mathbf{w}}(s_{t+1})$
8:         $\boldsymbol{e} \leftarrow \gamma\lambda\boldsymbol{e} + \nabla_{\mathbf{w}} v_{\mathbf{w}}(s)$
9:         $\mathbf{w} \leftarrow \mathbf{w} + \alpha\delta\boldsymbol{e}$
10:        **for** $s_k$ in $\mathcal{N}(s_t)$ **do**
11:            take gradient step wrt $\mathbf{w}$ minising $\ell_{sd} = \big(\eta v_{\mathbf{w}}(s_t) + (1-\eta)v_{\mathbf{w}}(s_k)\big) - v_{\mathbf{w}}(s_k)$
12:        **end for**
13:     **until** $s$ is terminal
14: **end for**
15: **Return** $\mathbf{w}$

**Algorithm 4** Structural Dueling Q learning +

1: initialise $\mathbf{w}$
2: initialise $\boldsymbol{e} = \mathbf{0}$
3: observe initial state $S$
4: pick action $A \sim \pi(q_{\mathbf{w}}(S))$
5: $v \leftarrow \max_a q_{\mathbf{w}}(S, a)$
6: $\gamma = 0$
7: **repeat**
8:     take action $A$, observe $R$, and $S'$
9:     $v' \leftarrow \max_a q_{\mathbf{w}}(S', a)$
10:     $\delta \leftarrow R + \gamma v'$
11:     take gradient step wrt $\mathbf{w}_q$ minising $\ell_{td} = \delta - q_{\mathbf{w}}(S, A)$
12:     **repeat** for all $s_k \in \mathcal{N}(s_t)$
13:         take gradient step wrt $\mathbf{w}_v$ minising $\ell_{sd} = \big(\eta v_{\mathbf{w}}(s_t) + (1-\eta)v_{\mathbf{w}}(s_k)\big) - v_{\mathbf{w}}(s_k)$
14:     **until**
15: **until** done

## C Models Hyperparameters

STRUCTURAL DUELING DQN

| | |
|---|---|
| exploration strategy | $\epsilon$-greedy with linearly decaying epsilon from 1. to 0.01 in first 10% steps |
| maximum number of steps per episode | 10,000 |
| number of hidden units in dense layers | 512 |
| number of dense layers in torso | 3 |
| number of dense layer in dueling heads | 1 |
| replay capacity | 10,000 |
| batch size | 32 |
| update target network frequency | 400 steps |
| optimiser | RMS prop |
| initial learning rate (temporal updates) | 0.0001 |
| initial learning rate (structural updates) | 0.01 |

## D    CONVERGENCE OF STRUCTURED VALUE ITERATION

The convergence argument for value iteration demonstrates first that policy evaluation under the Bellman Equation is a contraction mapping, meaning that its repeated iteration will converges to a single point, and that this point is the value function under the current policy. It then demonstrates that the outer loop of value iteration similarly converges to a single point, which is the optimal value function (Bellman, 1957; Vajjha et al., 2020).

### D.1    PROOF THAT POLICY EVALUATION CONVERGES

Like Bellman policy evaluation, we can show that policy evaluation with structural constraints is a contraction mapping (under some assumptions). First, we recall the proof that Bellman policy evaluation is contractive. A contraction mapping is any operator $F$ on a vector space $V$ with a norm $\|\cdot\|$ such that for some $\gamma \in (0, 1)$ and for all $v, u \in V$,

$$\|F(v) - F(u)\| \leq \gamma \|v - u\| \tag{9}$$

For the Bellman Equation in a tabular environment, we consider two value functions that are vectors $v, u \in V$, where $V$ has dimensionality equal to the number of states. If the mapping $F_t$ is the Bellman equation (where the subscript $t$ is for temporal credit assignment), $F_t(v) = r^\pi + \gamma P^\pi v$, for reward vector $r^\pi$ and transition matrix $P^\pi$ under a given policy, we can write:

$$\|F_t(v) - F_t(u)\| = \|(r^\pi + \gamma P^\pi v) - (r^\pi + \gamma P^\pi u)\| = \|\gamma P^\pi(v - u)\| \tag{10}$$

We consider the $l^\infty$ norm $\|\cdot\|_\infty$, in which the norm of a vector is equal to the maximum absolute value of its entries. Furthermore, we know that each row of the transition matrix $P^\pi$ sums to 1, such that $P^\pi \mathbb{1} = \mathbb{1}$. We then recall:

$$\|F(v) - F(u)\| = \|\gamma P^\pi(v - u)\| \leq \|\gamma P^\pi \mathbb{1}\| v - u\|\| = \|\gamma \mathbb{1}\| v - u\|\| = \gamma \|v - u\| \tag{11}$$

Now, we turn to structured updates. We constrain our structured neighbours to have the same value as each other, meaning that values for neighbour states $s_i$ are interchangeable with that of its neighbours, $s_j \in \mathcal{N}(s_i)$. One way of saying this is that for any combination of weights $w_j$ such that $\sum_j w_j = 1$, $V(s_i) = \sum_{s_j \in N} w_j V(s_j)$. We can write this in linear algebra form as $v = Wv$, where $W$ is a non-negative matrix of weights in which $W_{ij} > 0$ only if $s_j$ is a neighbor of $s_i$ and $W\mathbb{1} = \mathbb{1}$ (we assume each state can be its own structured neighbor). The Bellman operator $F_t(v) = r^\pi + \gamma P^\pi v$ can therefore be expanded to $F_s(v) = W(r^\pi + \gamma P^\pi v)$, where $W$ denotes any valid broadcasting of reward and value information to structural neighbors. Following the previous steps from the previous proof:

$$\begin{aligned}
\|F_s(v) - F_s(u)\| &= \|W(r^\pi + \gamma P^\pi v) - W(r^\pi + \gamma P^\pi u)\| \\
&= \|\gamma W P^\pi(v - u)\| \leq \|\gamma W P^\pi \mathbb{1}\| v - u\|\| \\
&= \|\gamma W \mathbb{1}\| v - u\|\| \\
&= \|\gamma \mathbb{1}\| v - u\|\| \\
&= \gamma \|v - u\|
\end{aligned}$$

This proves that VI and Structural VI converge to a single point.

### D.2    PROOF THAT POLICY EVALUATION CONVERGES TO THE CORRECT VALUE FUNCTION

The proof that the point at which VI converges is the actual value function $v^\pi$ is given by the fact that $v^\pi$ is a fixed point of the Bellman equation, because $v^\pi = r^\pi + P^\pi v^\pi$. Generalizing this step to the case of structured updates requires us to make a further assumption. Under the optimal policy $\pi^*$, we know that $v^* = W(r^{\pi^*} + P^{\pi^*} v^*)$; by construction, the *optimal* value of all structured neighbors are equal.

However, it may not be the case that the value of all structured neighbors are equal under *every* policy. Consider an environment with two rewarded ends of a linear track MDP: Each state will share value with their symmetric counterparts on the other side of the track, but not under a policy in which the agent moves away from one end and towards the other.

It is not clear how likely such a policy is to be encountered during learning, especially early on when the network is close to its random initialization. Because we only constrain *value* to be transformation invariant, but do not explicitly constrain *policy* to be equivariant, the value function, not the policy, is explicitly forced to respect the symmetry constraints. As the value function learns, and value is shared among neighbours, affecting policy, this assumption is increasingly valid.

Our empirical observation is that the assumption suffices for performance and sufficiency gains. However, the extent to which the policies (modulo the equivariant transformation) differ may contribute to the convergence time and stability, and perhaps suggest the optimal value of $\eta$. We could also enforce policy equivariance among our structured neighbors, which might further improve performance and provide additional theoretical guarantees, but would require additional label information to be provided and hard-coded into the agent architecture. We consider an exploration of these considerations an important area for future work.

### D.3 PROOF VALUE ITERATION CONVERGES TO THE OPTIMAL VALUE FUNCTION

That value iteration provably converges to the optimal policy will hold in the case of structured updates as well, given the same assumptions as above. The proof that standard value iteration converges to the optimal value function has a similar form to the proof that policy evaluation converges to the current value function. The operator $F_{\text{vi}}$ is given by the Bellman optimality operator $(F_{\text{vi}}(q))(s) = r(s) + \gamma \max_a q(s,a)$ (Equation 2). We can show that $F_{\text{vi}}$ is also a contraction mapping with Q-functions $q_v$ and $q_u$ (corresponding to state value functions $v$ and $u$):

$$\|(F_{\text{vi}}v)(s) - (F_{\text{vi}}u)(s)\| = \|\gamma(\max_a q_v(s,a) - \max_a q_u(s,a))\|$$
$$\leq \gamma \|\max_a q_v(s,a) - \max_a q_u(s,a)\|$$
$$\leq \gamma \max_a \|q_v(s,a) - q_u(s,a)\|$$
$$\leq \gamma \|v(s) - u(s)\|$$

Thus, $\|F_{\text{vi}}v - F_{\text{vi}}u\| \leq \gamma \|v - u\|$. Thus, this operator will contract to a point, and the point will be that at which taking the maximum value over available actions does not change the value, which is necessarily the optimal value function.

Replacing $v$ with a structurally updated $Wv$ allows $F_{\text{vi},s}$ to remain a contraction operator:

$$\|F_{\text{vi},s}v - F_{\text{vi},s}u\| \leq \gamma \|v - u\|$$
$$= \gamma \|W\mathbb{1}\|\|(v - u)\|\|\|$$
$$= \gamma \|\mathbb{1}\|\|(v - u)\|\|\|$$
$$= \gamma \|(v - u)\|$$

And by construction, since $v^* = Wv^*$ at the optimal value function $v^*$, the convergence point should be the same. As noted above, this requires the assumption that the value of each state is equal to that of its structured neighbors, which is only true at the optimal value function. However, the optimal value function is the point of convergence, meaning structured updates should not keep value iteration from converging to the optimal value function.

