# OpenReview forum: "Beyond Temporal Credit Assignment in Reinforcement Learning"
_ICLR.cc/2023/Workshop/RRL — RRL 2023 Poster_

### Official Review · Reviewer_jnEw · 2023-02-28
**An interesting work, but not within the scope of Reincarnating RL.**

**Rating:** 1
**Confidence:** 4

**Review:**

In this paper the authors highlight the inefficacy of temporal credit assignment in Reinforcement Learning, as it induces slow learning, besides not scaling well to large and complex environments. They argue that this failure, whether with or without function approximation, would stem from the lack of explicit consideration of structural information on the state space of the environment being dealt with. For instance, for a given task, the structure of the environment might induce the existence of sets of equivalent/similar states in the sense that they would carry the same value. The authors call these structural neighbours.

Posterior to a formal introduction to the notion of structural neighbours, the authors proceed with proposing structural variants of standard RL optimisation algorithms (Dyna, TD & Duelling DQN), by augmenting those with a structural update which supplements the temporal propagation of the information provided by the reward with another propagation amongst sets of structural neighbours.

The resulting algorithms are then evaluated against a variant of the four rooms domain, the game of Solitaire and its simplified version MiniSolitaire. In all instances, experiments demonstrate that the approach performs faster convergence than the standard (solely) temporal counterpart. In the four rooms environment, a map of the evaluate value function shows that it converges to the expected value map in the case of structural Dyna while standard Dyna remains far from it ; in the case of the Solitaire & MiniSolitaire environments, the structural DQN reaches the maximum total return much faster than its nonstructural counterpart.

-----

Despite the interesting comment on the inefficacy of credit assignment in standard RL methods, the demonstrated value of the proposed structural framework, and although this provides not only proof of concept but relevant contribution to the field of RL in general, we, unfortunately, fail to see how the current work fits within the scope of Reincarnating RL. We believe this work would be more suitable for venues (workshop or main conference track) centred on topics such as credit assignment, RL optimisation, ...etc

-----

We thank the authors for the good read, and hope this review will provide useful feedback.

---

### Official Review · Reviewer_E271 · 2023-03-01

**Rating:** 2
**Confidence:** 3

**Review:**

The paper aims to make use of prior knowledge in RL, specifically in the case where it is known that certain sets of states are functionally equivalent and should have the same value. In addition to normal TD learning, it proposes updating states' value based on states that are known to be in the same functionally equivalent set, enabling faster learning.

First, I'm not fully convinced about the novelty of this approach compared to the rich literature on works like bisimulation state abstractions, successor representations, etc. Second, my sense is that in practice, especially with realistic state/action spaces, it will be quite difficult for a human user to specify these invariances and know this set of equivalent states. Also, if the human does know this, I wonder if there are better ways of incorporating the prior knowledge, for instance, shouldn't the human then design a state abstraction that removes this ambiguity? It makes sense that this method works on Solitaire, but I'm not totally convinced there are many other domains where such a method would be practical to use.